# Alginate Ag for Composite Hollow Fiber Membrane: Formation and Ethylene/Ethane Gas Mixture Separation

**DOI:** 10.3390/membranes12111090

**Published:** 2022-11-02

**Authors:** Evgenia Dmitrieva, Evgenia Grushevenko, Daria Razlataya, George Golubev, Tatiana Rokhmanka, Tatyana Anokhina, Stepan Bazhenov

**Affiliations:** A.V. Topchiev Institute of Petrochemical Synthesis RAS, Leninsky Pr. 29, 119991 Moscow, Russia

**Keywords:** silver alginate, composite membrane, gas separation properties, ethylene–ethane separation

## Abstract

Membranes based on natural polymers, in particular alginate, are of great interest for various separation tasks. In particular, the possibility of introducing silver ions during the crosslinking of sodium alginate makes it possible to obtain a membrane with an active olefin transporter. In this work, the creation of a hollow fiber composite membrane with a selective layer of silver alginate is proposed for the first time. The approach to obtaining silver alginate is presented in detail, and its sorption and transport properties are also studied. It is worth noting the increased selectivity of the material for the ethylene/ethane mixture (more than 100). A technique for obtaining a hollow fiber membrane from silver alginate has been developed, and its separating characteristics have been determined. It is shown that in thin layers, silver alginate retains high values of selectivity for the ethylene/ethane gas pair. The obtained gas transport properties demonstrate the high potential of using membranes based on silver alginate for the separation of an olefin/paraffin mixture.

## 1. Introduction

Ethylene and propylene are large-tonnage raw materials for the petrochemical industry [1]. Global demand for ethylene exceeds 150 million tons per year, and it is believed that demand is increasing annually by 3% [2]. The main method of obtaining ethylene is the steam cracking of ethane, propane or naphtha, which inevitably leads to the formation of a mixture of products ethane–ethylene [3]. The high energy consumption of cryogenic distillation [4] used for the separation of this gas mixture significantly increases the cost of the final product. As a result, the affinity of boiling temperatures (t_boil_ = −89 °C and −103 °C for ethane and ethylene, respectively) leads to a necessity to carry out distillation in rectification columns with more than 100 plates working at temperature around −25 °C and pressure > 2000 kPa [5]. According to the literature, the cost of such separation of olefins and paraffins is up to $5 billion per year [6]. All of this necessitates the search for and development of other methods of gas separation, which would reduce the cost of materials, energy and, ultimately, the cost of ethylene.

Among the alternative techniques of gas separation, membrane processes are of the greatest interest due to their low energy consumption, automatic functioning, and ease of integration into the technological process [7]. Membrane separation processes are considered both as independent methods and working in tandem with distillation processes [8]. The latter approach allows a 30% reduction in energy consumption for separation [9]. For the separation of olefins and paraffins membranes from polymeric [10] and inorganic [11] materials are used.

It has been known since the 1970s that olefins can form complexes with silver ions [12,13]. This property is actively used in the manufacture of adsorption materials that selectively extract olefins [14,15]. For these purposes, silver-modified porous substrates based on carbon [16,17], zeolites [18], acidic aromatic Brensted frameworks [19], etc., are used. It is important that the property of adsorption of olefins by silver-containing materials can also be used in membrane separation processes. Thus, the membranes containing silver facilitate the transport of unsaturated hydrocarbons [20], which is more efficient compared to the molecular sieve and diffusion–sorption transport mechanisms (Figure 1), which significantly increases the permeability of such membranes for the target olefins, and hence their selectivity [21].

In gas separation silver-containing membranes of perfluorosulfonic acid cation-exchange membranes MF-4SK [22], poly-2-ethyl-2-oxazoline [23,24], polyetheramine [7,25], polyether blockamide Pebax [26,27] and others are known. Most silver-containing membranes require wet gas, which combines well with steam cracking. In particular, a cross-linked poly(ethylene glycol) diacrylate membrane and silver bis(trifluoromethylsulfonyl)imide salt was developed that is highly stable in reducing gas environments [28]. The incorporation of silver in membranes is a common practice used not only in gas separation but also in ultra- [29], nanofiltration [30], and pervaporation processes [31], where silver contributes to increase membrane permeability [32], selectivity [33], mechanical strength [34], resistance to contamination [35] and antibacterial activity [36]. Sometimes the ability of some silver-containing liquids to extract olefins is used in gas separation [37]. In this case silver is not included in the membrane matrix but is introduced in the form of impregnating liquids [38,39].

Another promising method for the separation of olefins and paraffins is application of gas-liquid contactors [40,41]. Separation in this process is provided by different solubility of olefins and paraffins in absorbent liquid circulating on one side of the membrane [42]. When separating olefins and paraffins the most common absorbents are silver nitrate [43] and silver tetrafluoroborates [44] in water and in ionic liquids [45]. Flat sheet membranes [46] can be used in membrane contactors, but hollow-fiber membranes [47] are preferred. This type of membrane has higher efficiency than flat membranes due to the larger membrane surface area and compactness [42]. The advantage of gas–liquid contactors is their high efficiency due to the combination of membrane and absorption technologies and the lack of the need to moisten the gas when working with materials susceptible to moisture [48,49,50,51]. Thus, circulation of the absorbent—an aqueous solution of silver nitrate—allows the material of the membrane selective layer to be kept wet and constantly replenish the loss of silver ions, ensuring the stability of the membrane operation over time.

A new and interesting direction is the creation of membranes for gas–liquid contactors with hydrophilic selective layers from natural materials. In this regard, a patent on gas separation [52] using membranes impregnated with silver nitrate based on polyimide PI-5, the pores of which are filled with hydrophilic sodium alginate/chitosan, has attracted attention. The hydrophilic polymers are applied to the porous substrate by dropwise application from solutions with concentrations of 2000–3000 ppm; then, the membranes are immersed for 2.5–3 h in 1.5–3 M solutions of silver nitrate. The obtained membranes are characterized by lower permeabilities due to the formation of additional resistance to mass transfer, but their selectivities are several orders of magnitude higher compared to untreated membranes. This work demonstrates the positive effect of using not only silver cations, but also of natural polysaccharides. Thus, impregnation of pores with chitosan and sodium alginate allowed for increasing propylene/propane selectivity from 3.9 to 203–315. Gas separation, in this case, was carried out using wet gases, which causes some technological difficulties, which could be circumvented by changing the membrane process from conventional gas separation to the use of membranes in gas–liquid contactors. In this work, the hydrophilic polysaccharides sodium alginate and chitosan were used only as pore fillers. However, the membranes consisting completely of these polymers are well known from literature [52,53,54].

Materials based on alginate salts with introduced silver are of increased interest in various fields. Thus, alginate with the introduction of silver and particles containing it is used to create a variety of medical products [55,56,57] and films for product packaging [58]. Sodium alginate is used for the synthesis of silver nanoparticles [59]. Due to its high viscosity, alginate prevents the coagulation of formed silver nanoparticles and maintains their high dispersion. Importantly, the introduction to alginate silver is known not only at the level of research works, but has also found commercial application in the form of wound dressings, known under the brand names “AskinCalgitrolAr” and “Alpe Alginate Ag Silver Dressing”.

Due to the presence of carboxyl and hydroxyl groups, alginate has a high density of negative charge, which contributes to the formation of strong bonds with silver ions by ionic and coordination mechanisms [60]. It is possible to prove the chelation of silver ions by hydroxyl and carboxyl groups of alginate through the disappearance of the 301 nm band originally present in silver nitrate solutions in the spectrum [61]. In this case, some silver ions are partially reduced to nanoparticles [62], which can be recognized by the appearance of peaks on the spectrum in the visible region around 400–450 nm [63].

The application of membranes based on alginate with inserted silver-containing additives is presented in a number of research works devoted to water treatment [59,64,65]. For example, membranes based on sodium alginate with embedded particles of olystyrene-4-sulfonic acid co maleic acid sodium salt capped silver nanoparticle (Ag_Np) were used in pervaporation [59]. Silver nanoparticles (AgNPs), which were synthesized with bovine serum albumin (BSA) and carboxymethyl chitosan (CMCS), were introduced into calcium alginate membranes to purify water from dyes and improve mechanical and transport properties [64]. Chitosan membranes modified with silver nanoparticles and alginate nanofibers cross-linked with calcium and glutaric aldehyde were used to purify water from oils [65].

The development of filtration membranes based on silver alginate seems to be a promising task for the separation of ethylene/ethane mixtures. The use of pure silver-cross-linked alginate in filtration processes has not been investigated so far. It is worth noting that sodium alginate has good film-forming properties and can be applied as a selective layer on the outer surface of hollow fibers [66,67,68], which is of great practical importance in terms of creating gas separation modules and gas–liquid membrane contactors. This allows alginate to be considered as a promising hydrophilic silver-containing coating for hollow fiber membranes.

The purpose of this work is to develop a composite hollow fiber membrane with a selective hydrophilic layer of silver alginate for separation of olefins and paraffins. Alginate cross-linked with silver cations will be used for the first time as a selective layer material. Silver cross-linking of the polymer will provide increased selectivity of the membranes for olefins and their facilitated transport.

## 2. Materials and Methods

### 2.1. Materials

Sodium alginate was purchased from Rhône-Poulenc (Paris, France) and used without further purification. The molecular weight of the polymer was Mw = 1.2 × 10^6^ [54], the ratio of β-D-mannuronic (M) and α-L-guluronic (G) acids equal to M/G = 5.5 [54]. A polyester nonwoven fabric manufactured by Crane Technical Materials (Boston, MA, USA), flat-sheet membranes and hollow fibers from polysulfone (PSF) were used as supports for composite membranes. The membranes were crosslinked using silver nitrate (95%, Chimmed, Podolsk, Russia).

As flat sheet porous support fluoroplastic porous membrane MFFK-1 (Vladipor, Vladomir, Russia) was taken. Polysulfone (PSF, BASF Ultrason^®^ S 6010, Heidelberg, Germany), solvent N-methylpyrrolidone (MP, AcrosOrganics, 99% extrapure, Geel, Belgium) and pore-forming additive polyethylene glycol (PEG-400, AcrosOrganics, Geel, Belgium) were used for formation hollow fiber support.

Gas transport properties were studied using nitrogen (OOO “Argon”, Moscow, Russia), carbon dioxide (NII KM, Moscow, Russia), helium (99.97%, NII KM, Moscow, Russia) and a mixture of ethylene–ethane gas. The gas mixture was prepared at Moscow gas processing plant (Moscow, Russia). The ethylene/ethane ratio was 20% vol./80% vol.

### 2.2. Characterization of Silver Alginate

IR spectra of sodium alginate powder were recorded in the ATR reflection mode on a HYPERION-2000 IR microscope coupled to a Bruker IFS-66v/s Fourier spectrometer (Elmsford, NY, USA) (scan 50, Ge crystal, resolution 2 cm^−1^, range 600–4000 cm^−1^).

The visible light spectrum was recorded in PE-5400UF spectrophotometer (PromEcoLab, Saint-Petersburg, Russia). The measurement frequency was 1 nm.

Registration of SERS spectra was carried out on a portable Romanov spectrometer “KhimExpert” TU 4434-018-29095820-13 (Moscow, Russia). The experimental samples were studied using a laser with a wavelength of *λ* = 532 nm in the wavenumber range from 359 to 4059 cm^−1^ at a focal length of 50 mm. Samples of alginates in the form of dried films 20 µm thick were subjected to analysis. The spectral resolution of the instrument was 15 cm^−1^. Measurement times were no more than 30 s.

Sorption measurements were conducted by placing membrane samples in water for several days after preliminary weighing of the samples. After soaking in water, the membranes were taken out, the excess water from the surface of the samples was removed with filter paper, and the membranes were weighed daily. The measurements were stopped when the mass of the membrane did not change for 2 days. The sorption (*K*) was calculated by the Formula (1):(1)$$K=\frac{m_{1}-m_{o}}{m_{0}}$$
where *m*_0_, *m*_1_ are the masses of the dry and swollen sample, respectively.

The sorption of ethane and ethylene on samples of calcium alginate and silver alginate was measured using an XEMIS-002 gravimetric sorption analyzer (HidenIsochema, Warrington, UK) at temperatures of 30 and 60 °C and pressure *P* = 1 bar. Before measurements, the samples were evacuated at 110 °C for 72 h to remove adsorbed water. Next, using the helium calibration method described in detail in [69], the skeletal density of the samples was determined. The time to reach the equilibrium value of sorption, according to its sorption, for all gases and samples did not exceed 20 min. The values of gas densities used in the work at different pressures and temperatures were taken from the NIST REFPROP V.9.1 database (accessed on: 22 July 2022). All isotherms were taken at least 3 times to control the convergence of the results. Between measurements, the sample was evacuated for 2 h at the experimental temperature.

To study the quantitative characteristics of crosslinking, the mass of silver cations that pass from the crosslinking solution into the polymer was estimated. To do this, a sample of sodium alginate films was immersed for a week in a low-concentration solution of silver alginate 0.25% wt. The concentration of silver cations in the solution was evaluated before and after crosslinking according to the Gay-Lussac titrimetric method [70]. The obtained numerical values were compared with the EDX data.

### 2.3. Fabrication of Flat-Sheet Composite Membranes with the Selective Layer from Alginate Silver

Silver alginate membranes were prepared by the «phase inversion» method. An aqueous composition of sodium alginate (3 wt.%) was cast onto a flat sheet porous membrane using a doctor blade with a gap thickness of 200 μm. The exposure time to the crosslinking agent—0.7 M solution silver nitrate in water—ranged from 1 to 80 min.

### 2.4. Fabrication of Hollow Fiber PSF Supports

For the preparation of hollow fiber PSF support used a 15% wt. solution of PSF in N-methylpyrrolidone with the addition of 30% wt. PEG (400). The solution was stirred at 50 °C for 24 h until it became homogeneous.

Samples of porous asymmetric hollow fiber support from PSF were obtained by the method of dry–wet phase inversion using the setup presented in our previous works [71,72]. In the process of forming hollow fibers, an annular die with an inner/outer diameter of 0.8/1.7 mm was used. To form an outer selective layer, distilled water was supplied to the outer surface of the hollow fiber. Distilled water was also used as a bore fluid. After molding, the samples of hollow fiber membranes were sequentially washed for three days with tap water, then with ethanol for 2 h, then with n-hexane for 2 h, after which they were dried in air at room temperature.

### 2.5. Fabrication of Hollow Fiber Composite Membranes with the Selective Layer from Alginate Silver

The selective layer on hollow fiber membranes was formed from aqueous solutions of sodium alginate 0.1–3% wt. Several methods have been tried for the formation selective layers from alginate silver on hollow fiber support.

(A) The hollow fiber PSF support was soldered into the measuring module, an aqueous solution of silver alginate was poured into the module. After 1 min it was replaced with a 0.7 M aqueous solution of silver nitrate. Crosslinking was carried out for 1 min.

(B) The hollow fiber PSF support was immersed for 1 min in an aqueous solution of sodium alginate, after which it was immediately soldered into the module. A 0.7 M aqueous solution of silver nitrate was poured into the module for 1 min.

(C) The hollow fiber PSF support was immersed for 1 min in an aqueous solution of silver alginate and then for 1 min in a 0.7 M aqueous solution of silver nitrate. The composite membranes were dried in air and sealed into a module.

### 2.6. Characterization of the Morphology of Composite Membranes with the Selective Layer from Alginate Silver

Scanning electron microscopy (SEM) was used to characterize the structure and morphology of the membranes. SEM was carried out on a Thermo Fisher Phenom XL G2 Desktop SEM (Waltham, MA USA), equipped with a module for energy-dispersive elemental spectroscopy (EDX). Cross-sections of the membranes were obtained in liquid nitrogen after preliminary impregnation of the specimens in isopropanol. A thin (5–10 nm) gold layer was deposited on the prepared samples in a vacuum chamber (~0.01 mbar) using a desktop magnetron sputter “Cressington 108 auto Sputter Coater” (Watford, UK). The accelerating voltage during image acquisition was 15 keV. Further images analysis and determination of the selective layer thickness was carried out using the Gwyddion software (ver. 2.53) (Czech Metrology Institute, Brno, Czech Republic).

### 2.7. Characterization of Gas Permeability of Composite Membranes with the Selective Layer from Alginate Silver

The study of the gas transport characteristics of flat-sheet and hollow fiber composite membranes was carried out using the equipment presented in our last articles [73,74]. The measurements were performed at a temperature of 25 °C and feed membrane pressure from 1 to 3 atm (gauge).

The pressure of the gas mixture fed to the membrane cell was set by the pressure regulator and detected by the pressure gauge. Permeate was degassed by a membrane vacuum pump Ilmvac 030 Z-EC (Ilmvac (Welch), Ilmenau, Germany) and its flow rate was measured by a soap film meter Flowmeter-P. Partial component fluxes were calculated based on their concentrations in the feed, which was determined by the gas chromatography technique. Retentate flow was regulated by a needle valve and measured by a soap film meter Flowmeter-P. The pressure of feed, retentate, and permeate was determined using feed, retentate, and permeate pressure gauges, respectively.

Stage cut is an important parameter to characterize membrane properties in the separation of multicomponent mixtures. According to IUPAC terminology, stage cut (*θ*) is a parameter defined as the fractional amount of the total feed entering a membrane module that passes through the membrane as permeate:(2)$$\theta =\frac{J^{perm}}{J^{feed}}$$
where *J^perm^*—volume permeate flow (cm^3^/s), *J^feed^*—volume feed flow (cm^3^/s).

In this work stage cut of 5% was adjusted by regulation of the retentate flow. As was noted in Reference [75], the low value of the stage cut allows the change in the concentration along the membrane cell to be neglected. In this work, a simplification was assumed that the upstream concentrations of components are equal to their average concentrations between the inlet and the outlet of the membrane cell. The composition of feed, permeate and retentate was analyzed employing gas chromatograph “Gazochrom-2000” (Chromatek, Yoshkar-ola, Russia). The chromatograph was equipped with a thermal conductivity detector and a nozzle chromatographic column packed with 20% wt. heptadecane on diatomite carrier. The analysis of 0.5 mL gas sample was carried out at the dosing valve thermostat temperature of 50 °C, columns thermostat temperature of 50 °C, detectors thermostat temperature of 160 °C, and the flow rate of carrier gas (helium) of 30 mL/min.

Membrane permeance was calculated using the formula:(3)$$P/l=\frac{C_{i}^{perm}\cdot J^{perm}}{S\cdot \left(C_{i}^{feed}\cdot p^{feed}-C_{i}^{perm}\cdot p^{perm}\right)}$$

$C_{i}^{perm}$—concentration of the component *i* in the permeate (% vol.); $C_{i}^{feed}$—concentration of the component *i* in the feed (% vol.); $p^{feed}$—feed pressure (cm Hg); $p^{perm}$—permeate pressure (cm Hg); $J^{perm}$—volume permeate flow (cm^3^/s); *S*—membrane area (cm^2^).

The separation selectivity between the *i* and *j* components was calculated using the formula:(4)$$\alpha_{mix}=\frac{{\left(P/l\right)}_{i}}{{\left(P/l\right)}_{j}}$$

### 2.8. Characterization of Water Permeability of Composite Membranes with the Selective Layer from Alginate Silver

The water permeability of the flat-sheet membranes was researched in dead-end cells with a transmembrane pressure of 5 bars, created with helium. The water permeability of the hollow fiber membranes was researched in flow cells with a transmembrane pressure of 1 bar, the scheme of which can be found in the article [76].

The permeate flow was determined by the gravimetric method. A liquid receiver was installed at the outlet of the cell. The mass of permeate passing through the membrane during the experiment was measured on a Sartorius laboratory balance with a measurement error of 0.001 g. Membrane performance was characterized by liquid permeance (*P*), which was calculated as follows:(5)$$P=\frac{m}{S\cdot \Delta t\cdot \Delta p}$$
where *m* is the mass of permeate (kg) passed through a membrane with an area *S* (m^2^) over a time Δ*t* (h), and Δ*p* is the pressure drop.

## 3. Results

### 3.1. Preparation of Silver Alginate Membranes

Obtaining highly selective membranes from silver alginate is an important multi-parameter problem for which it is necessary to determine silver amount into polymer. 

The first step was to estimate the minimum required mass of crosslinking silver nitrate per unit mass of sodium alginate polymer, which is important for the proper preparation of working solutions. Quantification of the silver consumed for crosslinking and its content in the polymer matrix allowed, in combination with data from various spectra, for some assumptions to be made about polymer crosslinking.

Assuming that cross-linking of alginate with silver involves complete replacement of sodium cations by silver cations, the theoretical concentration of silver (*C_Alg_*) in the polymer was calculated according to the following equations:(6)$$m_{Na}=23\cdot \frac{100}{M_{w}}\cdot \frac{M_{w}}{M_{Alg}}$$
(7)$$m_{Ag}=108\cdot \frac{100}{M_{w}}\cdot \frac{M_{w}}{M_{Alg}}$$
(8)$$C_{Ag}=\frac{m_{Ag}}{100-m_{Na}+m_{Ag}}\cdot 100\%$$
where 23 is the atomic mass of sodium; 100 is the accepted mass of sodium alginate (*m_Alg_*); *M_w_* is the molecular mass of the polymer; *M_Alg_* is the molecular mass of one alginate link; the ratio *M_w_/M_Alg_* represents the number of alginate links in the polymer, and 108 is the atomic mass of silver.

Substituting values *M_w_* = 1,200,000 g/mol and *M_Alg_* = 198 g/mol we obtain that the maximum concentration of silver in cross-linked polymer is *C_Ag_* = 38%. The consumption of silver nitrate (*Q_AgNO_*_3_) per unit weight of crosslinked sodium alginate is:$$Q_{AgNO3}=\frac{1}{M_{w}}\cdot \frac{M_{w}}{M_{Alg}}\cdot 170=\frac{170}{M_{Alg}}=0.85\;g/g$$
where the first multiplier represents the amount (mole) of polymer in 1 g of sample, the second multiplier is the number of alginate units in 1 mole of polymer, and 170 is the molar mass of silver nitrate.

Thus, for the complete crosslinking of each gram of sodium alginate, it is necessary to have at least 0.85 g of silver nitrate in crosslinking solution. Experimental data show that such a ratio is observed only when crosslinking the polymer with a low-concentration solution. In this work, a solution with a silver concentration 1.6 times higher than theoretically necessary for crosslinking was used for this purpose (with a mass calculation of 0.25% mass AgNO_3_ aqueous solution). Data of titrometric analysis of crosslinking solution before and after immersion in sodium alginate polymer suspension weighing 0.18 g for 1 week shows that the silver nitrate consumption was 0.80 g/g polymer, which is a fairly convergent value in comparison with the theoretical calculation. The EDX analysis showed a silver concentration of 31% in the polymer. Importantly, the EDX data show a uniform distribution of silver cations throughout the thickness of the crosslinked polymer (Figure 2).

Interestingly, the use of a highly concentrated solution of 0.7 M (10.6% wt.) silver nitrate leads to different results. Thus, the concentration of silver in the membranes cross-linked by highly concentrated solution for 1.5 to 80 min was 47.8–54.5% wt.; that is significantly higher than the maximum theoretical possible content of silver equal to 38%. At the same time, an increase of the cross-linking time up to 30 min leads to an increase of silver concentration, to visual “densification” of the film and decrease of water sorption values for dried samples (Figure 3). It is worth noting that cross-linking less than 1.5 min leads to the formation of films subsequently strongly swelling in water, losing their shape and mechanical strength.

The concentration of silver in the polymer exceeding the theoretically calculated 38% wt. means that in addition to the stoichiometric substitution of sodium cations for silver cations there is an additional introduction of silver into the polymer. The mechanism of this excessive “introduction” of silver cations can be explained by a number of spectra (Figure 4A,B).

Since polysaccharides are soft reducing agents [77], there is a gradual transformation of silver cations into its atoms. The atoms, grouped with each other, form nanoparticles, which can be detected by the appearance of peaks in the visible spectrum in the region of 400–450 nm [60]. In the alginate samples obtained in this work, the peak is at 425 nm (Figure 4A), which indicates the presence of silver nanoparticles with sizes larger than 10 nm in the polymer [63]. Another piece of evidence for the formation of silver nanoparticles is the appearance of a double-humped peak (Figure 4B) in the 1500 cm^−1^ region of the Raman spectrum [78]. It is known that the reduction of silver cations is always accompanied by darkening of solutions/samples, which is confirmed by the appearance of the obtained alginate films. Silver nanoparticles are clusters of atoms (Figure 5) due to which it becomes possible to introduce more silver than is possible according to the stoichiometric law.

At the same time, the reduction of silver cations does not occur completely, as washed and dried films show qualitative reactions to silver cations, causing the appearance of white and dark plaque on the surface when they are immersed in sodium chloride and sodium hydroxide, respectively. Thus, it can be argued that sodium alginate turns out to be cross-linked by both cations and silver nanoparticles simultaneously. The formation of “cross-linking” bridges in the polymer is possible due to the formation of ionic bonds, as well as chelates by hydroxyl groups of alginate and d-orbitals of silver, which has already been revealed by the example of another polysaccharide—chitosan [79]. Both of those Ag forms (nanoparticles and cation) show activity and selectivity in olefine–paraffin separation [80,81].

The obtained polymeric material has strong prospects for use in the separation of ethane and ethylene, because even in the dried form, it has selectivity in the sorption of unsaturated hydrocarbons (Figure 6B). Initial sodium alginate has less sorption of ethylene than silver alginate (Figure 6A). The main role in the obtained selectivity is played by silver cations and nanoparticles involved in cross-linking of the polymer. The same effect for silver nanoparticles in the membrane polymer was shown in work [82].

Figure 6 shows that the sorption of ethane by silver alginate and sodium alginate is the same and is equal to about 0.01 mmol/g, which indicates that this sorption is due to the properties of alginate itself. At the same time, sorption of ethylene differs significantly. Silver alginate has the maximum sorption on ethylene at 60 °C. In this case the ethylene/ethane selectivity at 1000 mbar is practically 5.

### 3.2. Development of Methods of Creation of Hollow Fiber Composite Membranes with a Selective Layer of Silver Alginate

The data obtained on the properties of silver alginate and its sorption selectivity for ethane/ethylene make the creation of membranes based on it for the separation of these gases promising. Hollow-fiber membranes are considered to be more efficient in membrane separation than flat membranes due to their high surface area. In this regard, the next important step in the study is to develop methods of applying selective layers on hollow fibers.

In the experimental phase, three methods of applying a thin selective layer of silver alginate to the outer surface of hollow polysulfone fibers are described. All of them are not equivalent and represent only different application attempts, from which one of the most optimal methods was chosen.

The application of silver alginate directly into the modules was unsuccessful because of the high viscosity of the aqueous sodium alginate solution (η_1% AlgNa_ = 98.2 mPa∙s) and the formation of a polymer layer on the inner surface of the module and/or polymer “plugs” (Figure 7B). Applying alginate to the fiber and sealing it into the module in undried form also did not lead to positive results. This method was unacceptable due to frequent contact of the fiber with the module walls and the occurrence of defects in the selective layer (Figure 7C). Successful and uniform application of silver alginate was only possible using the third method (Figure 7D). In this case, the selective layer was applied by dipping the hollow fiber sequentially in solutions of sodium alginate and silver nitrate. After that, the composite membrane was dried at room temperature and glued into the module.

The application of silver alginate by dipping with intermediate drying before pasting the fiber into the module yields composite membranes with an even selective layer. Application of silver alginate is controlled visually by the change in coloration of the fiber surface from white to brown. It is possible to control and change the thickness of the selective layer by using solutions of sodium alginate with different concentrations and different crosslinking times of the polymer in silver nitrate (Figure 8). Thus, the minimum obtained thickness of the selective layer is 2.3 µm with the application of one layer of 1% sodium alginate and a crosslinking time of 5 s. Increasing the concentration of alginate to 3% and the crosslinking time to 1 min leads to achieving a thickness of the selective layer of 11.2 μm.

Increasing the amount of sequential dipping of the fiber in solutions of sodium alginate and silver nitrate also leads to an increase in the thickness of the selective layer (Figure 9). No intermediate drying was performed between the application of different layers.

The use of lower concentrated alginate solutions, such as 0.1%, did not result in the appearance of a selective layer on the fiber surface (Figure 10).

Thus, the application of selective layers of silver alginate to hollow fibers was perfected. It was found that application of a selective layer is desirable on separate fibers, and before the creation of measuring modules, they should be subjected to partial drying. In addition, it was shown that increasing the thickness of the selective layer is associated with increasing the concentration of the polymer solution and its viscosity, increasing the cross-linking time in silver nitrate, and increasing the number of layers applied. Varying these parameters allowed the thickness of the selective layers to vary from 2.3 to 47.2 μm in this work.

### 3.3. Filtration Properties of Composite Membranes with Silver Alginate Selective Layer

In the course of the study, we succeeded in creating composite membranes with a silver alginate selective layer of two types: flat and hollow-fiber membranes. In both cases the selective layer turned out to be defect-free, which was confirmed by zero water flux (no water drop) from the dried membranes at pressure of 1.2 atm (Figure 11). 

Despite the fact that composite membranes with a selective layer of silver alginate are impermeable to water, they do not create a barrier to gases. At the same time, for both types of membranes there is a decrease in permeability in the series of the following gases: carbon dioxide–nitrogen–helium (Figure 12).

It is interesting that carbon dioxide has the highest permeability in the raw of the investigated pure gases. Moreover, CO_2_ has highest molecular mass and a kinetic molecular diameter close to nitrogen (3.3 A < 3.6 A). This may be due to the chemical interaction of carbon dioxide with silver cations due to the presence of free electron pairs of oxygen and free d-orbitals of silver cations. A similar mechanism of interaction is typical for silver cations and unsaturated hydrocarbons—alkenes—for which there is also an increased permeability for flat composite membranes with a selective layer of silver alginate (Table 1).

Table 1 shows that membranes with a silver alginate selective layer have permeability to alkenes 60–100 times higher than permeability to alkanes. This makes it possible to use such membranes for the enrichment of mixtures by alkenes. By increasing the pressure of the separated mixture, a growth in selectivity of separation is observed with a corresponding decrease of permeability that, apparently, is connected with peculiarities of the mechanism of ethylene transfer by silver ions and particles. The sorption of ethylene essentially depends on pressure. Thus, on the basis of data of Figure 5, the sorption of ethylene in silver alginate at 200–800 mbar differs almost 5 times. It should be noted that sorption growth for ethane is insignificant in the presented range. Thus, it is the sorption interaction that determines the increase of the selectivity of separation of the ethylene/ethane mixture. The decrease in the permeability coefficient can be related both to the compaction of swollen silver alginate under overpressure and to the complication of diffusion transfer as a result of the sorption interaction of silver nanoparticles and silver alginate with ethylene.

## 4. Conclusions

In the presented article a composite hollow fiber membrane with a selective layer of silver alginate and increased selectivity of ethylene release was obtained for the first time. A method for cross-linking sodium alginate with silver nitrate was developed, on the basis of which a method for applying a thin selective coating on a hollow fiber substrate of polysulfone has been proposed. Using the methods of Romanov and optical spectrometry and titrometry, the introduction of silver into the alginate matrix both in the form of nanoparticles and in the form of ions was confirmed. Comparison of the sorption of ethane and ethylene by calcium alginate and silver showed that the introduction of silver into the alginate matrix makes it possible to fundamentally increase the selectivity of sorption. Thus, for calcium alginate there is no sorption selectivity for ethylene/ethane vapor, and for silver alginate it reaches 5. Thus, the activity of the ions and silver nanoparticles introduced into alginate in ethylene transfer was confirmed.

The gas transport properties of the obtained membranes were investigated both for individual gases and for the ethylene/ethane mixture. It was shown that despite the low gas permeability of the membranes for individual gases, the permeability of ethylene in the mixture reaches more than 190 GPU. The ethylene/ethane mixture selectivity increased from 60 to 110 with a pressure increase from 140 to 200 kPa. Thus, composite membranes with silver alginate selective layers are promising for the separation of alkane and alkenes mixtures in gas separation processes. Particularly important is that such membranes are capable of handling dry gases. In addition, due to the water resistance of dried membranes based on silver alginate, it is possible to consider their use for the separation of alkanes and alkenes with gas–liquid contactors. The absorption liquid in the form of silver nitrate in such processes will perform a dual role: to contribute to the extraction of alkenes, as well as to “lend” the membrane over time, replenishing its losses in silver cations.

## Figures and Tables

**Figure 1 membranes-12-01090-f001:**
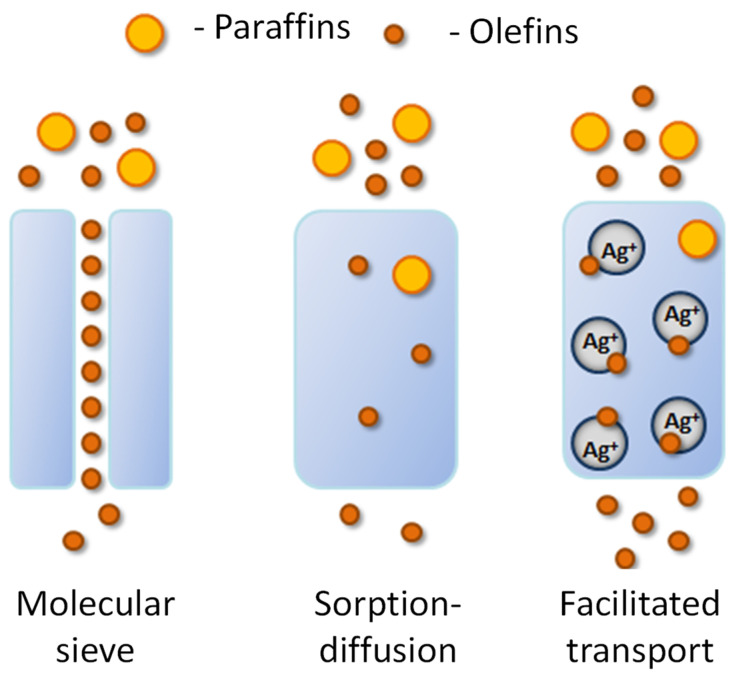
Schematic representation of the various transport mechanisms.

**Figure 2 membranes-12-01090-f002:**
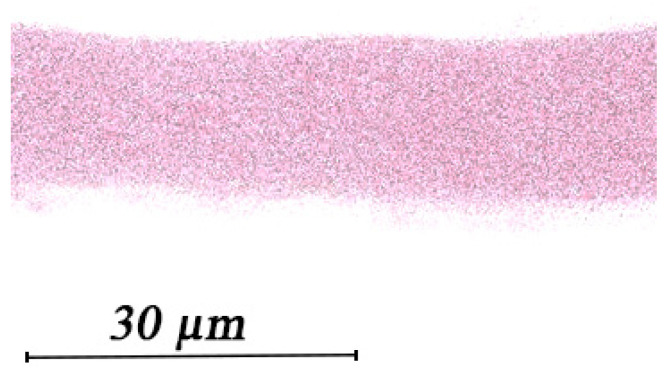
Silver distribution along the thickness of cross-linked alginate.

**Figure 3 membranes-12-01090-f003:**
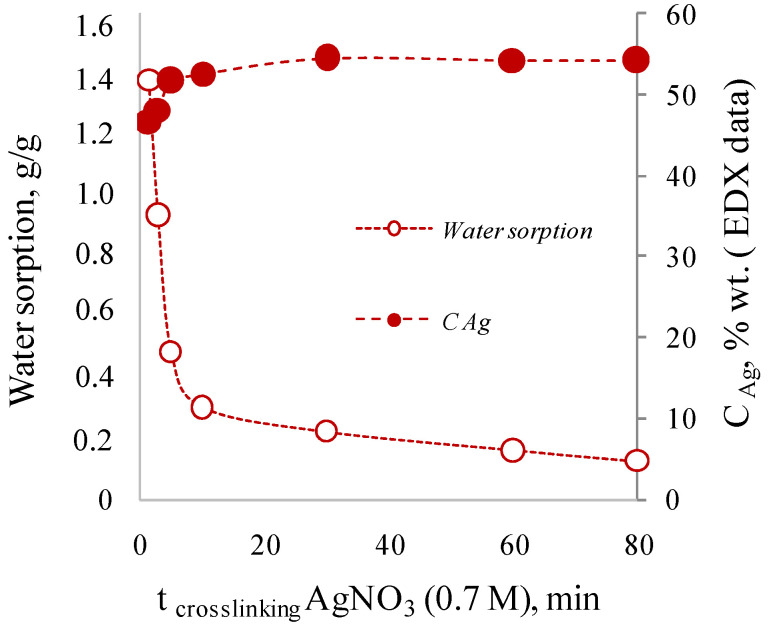
Effect of cross-linking time with 0.7 M AgNO_3_ on silver concentration in the cross-linked alginate matrix and on the sorption of water by the dried films.

**Figure 4 membranes-12-01090-f004:**
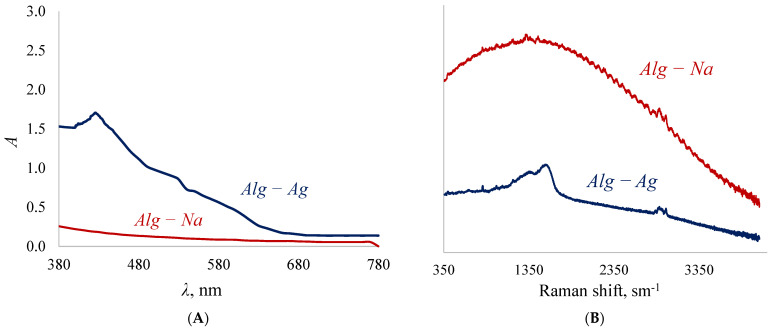
Visible (**A**) and Raman (**B**) spectra of sodium alginate and silver alginate.

**Figure 5 membranes-12-01090-f005:**
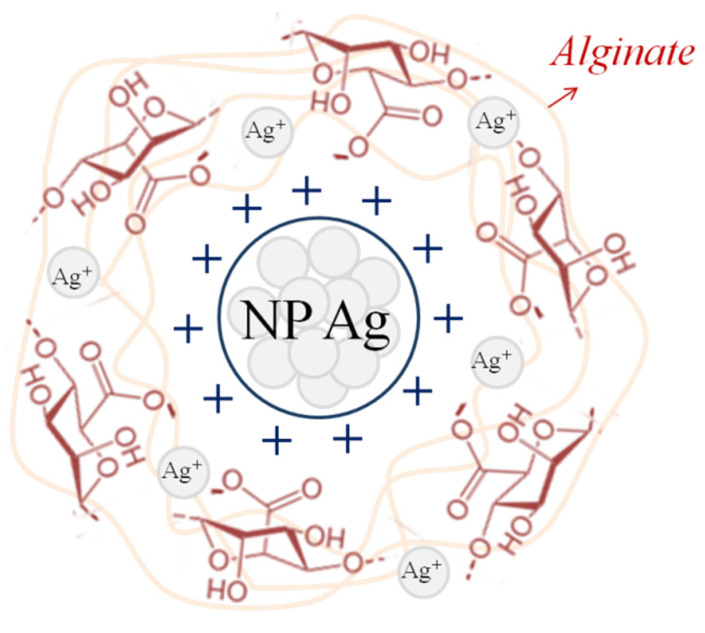
Schematic representation of the crosslinked polymer by nanoparticles (NP) Ag.

**Figure 6 membranes-12-01090-f006:**
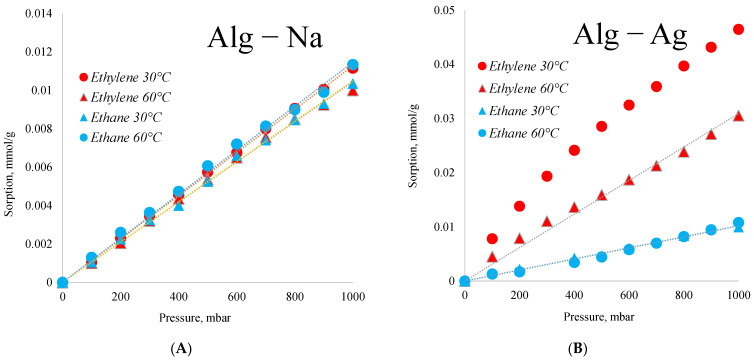
Sorption of ethane and ethylene by sodium alginate (**A**) and silver alginate (**B**) samples.

**Figure 7 membranes-12-01090-f007:**
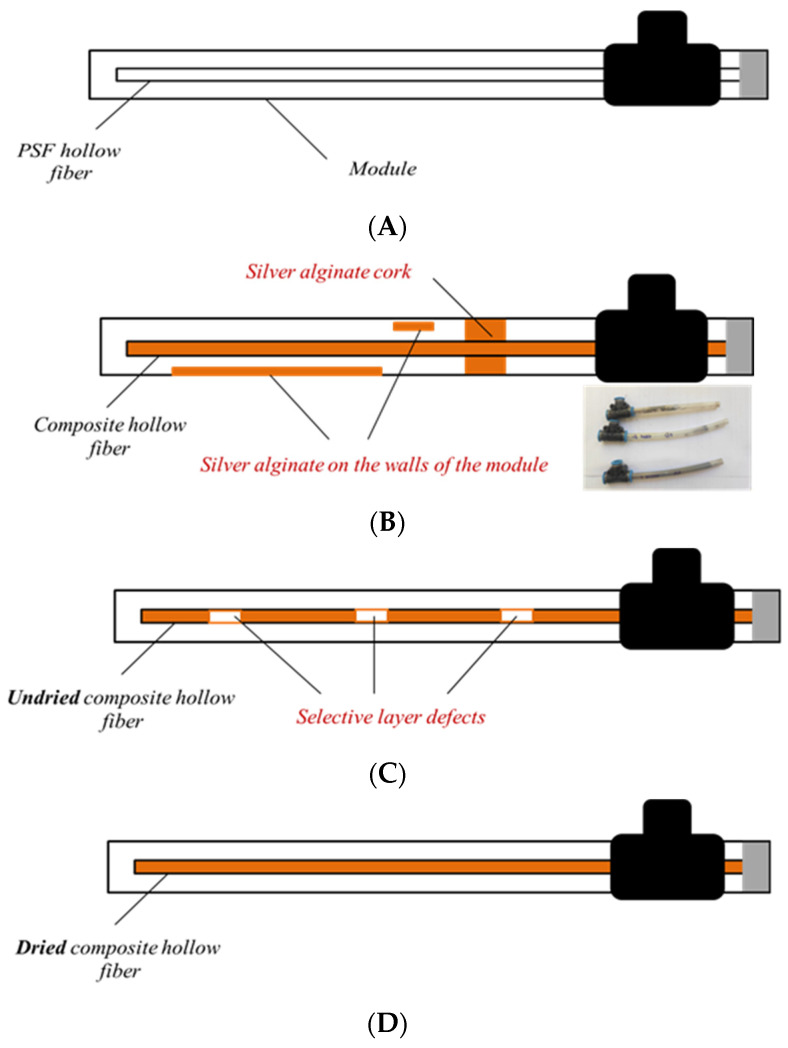
Schematic representation of the appearance of composite hollow fiber membranes and the problems encountered during their creation by different methods: (**A**)—initial module with hollow fiber porous support, (**B**)—composite hollow fiber silver alginate membrane with silver alginate cork and silver alginate into the module walls, (**C**)—defect composite hollow fiber silver alginate membrane, (**D**)—defect free composite hollow fiber silver alginate membrane.

**Figure 8 membranes-12-01090-f008:**
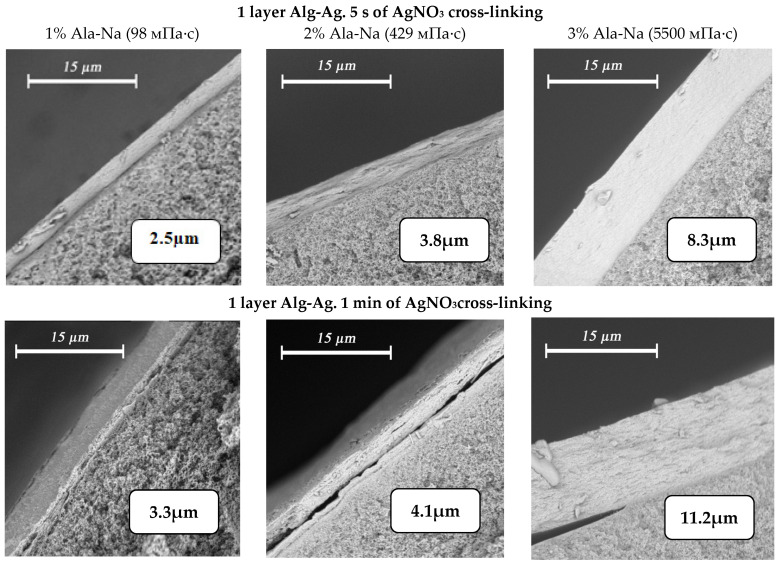
Variation of selective silver alginate layer thickness when varying the alginate concentration (1% wt.—first column, 2% wt.—second column, 3% wt. third column) and crosslinking time of the polymer (one layer—5 s (first row) and 1 min (second row)).

**Figure 9 membranes-12-01090-f009:**
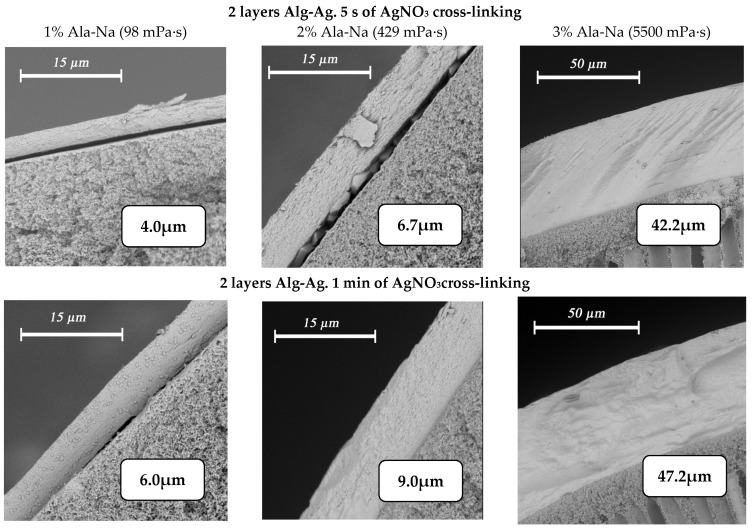
Changes in the thickness of the silver alginate selective layer when varying the alginate concentration (1% wt.—first column, 2% wt.—second column, 3% wt. third column)and crosslinking time of the polymer (two layers—5 s (first row) and 1 min (second row)).

**Figure 10 membranes-12-01090-f010:**
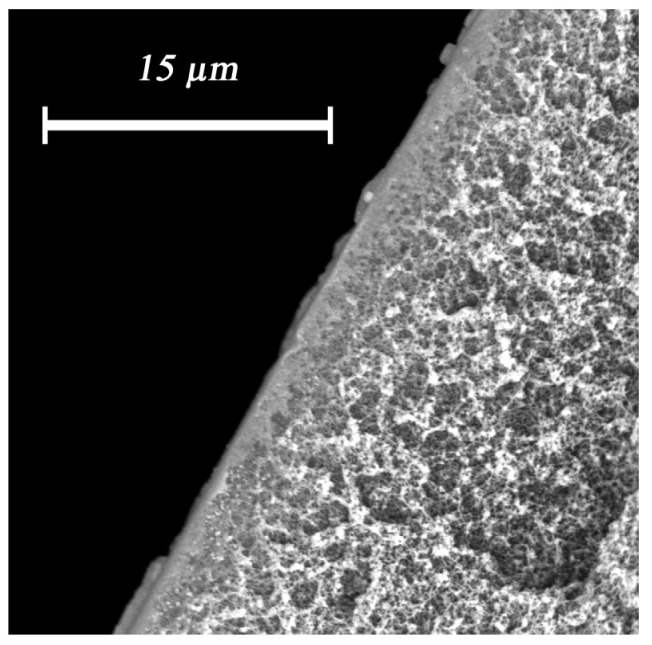
The absence of a polymer selective layer when using 0.1% sodium alginate solution using 0.1% sodium alginate solution.

**Figure 11 membranes-12-01090-f011:**
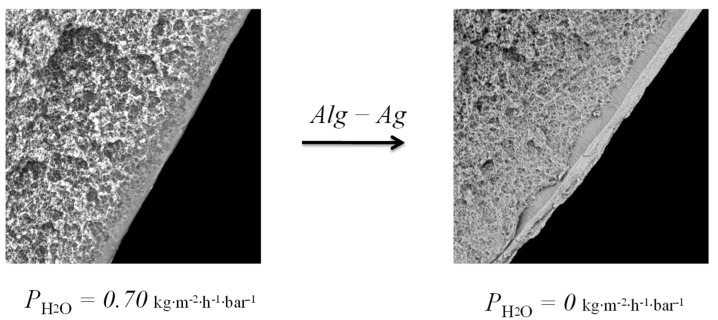
Effect of a selective layer of silver alginate on water flux.

**Figure 12 membranes-12-01090-f012:**
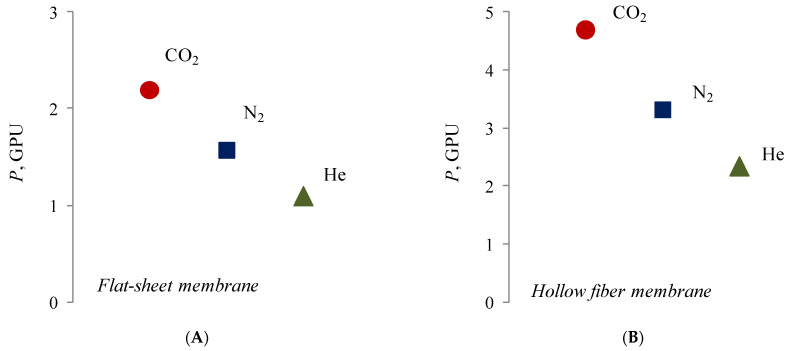
Permeability of flat (**A**) and hollow-fiber (**B**) composite membranes with a silver alginate selective layer for pure gases. Thickness of Alg-Agis 20 µm.

**Table 1 membranes-12-01090-t001:** Gas permeability of hollow fiber composite membranes over a mixture of ethane and ethylene at different pressures (selection fraction—4%, ethylene/ethane mixture composition—20/80% vol., permeate pressure—5 kPa. A selective layer was formed from 3% AlgNa, crosslinked by AgNO_3_ for 5 s and has a thickness 8.3 µm).

Pressure above Membrane, kPa	140	180	200
*P*_ethane_, GPU	3.2	2.4	1.2
*P*_ethylene_, GPU	192.7	168.7	132.5
α_mix_	60	70	110

## Data Availability

Not applicable.

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
