# Peer review of "Alginate Ag for Composite Hollow Fiber Membrane: Formation and Ethylene/Ethane Gas Mixture Separation"

_membranes, 2022, doi:10.3390/membranes12111090_

Round 1
Reviewer 1 Report
The development of advanced membrane materials for ethylene/ethane gas mixture separation is a very hot topic currently. In this work, alginate cross-linked with silver cations for the first time was used as a selective layer material. Silver cross-linking increased selectivity of the membranes for olefins. I would like to recommend its publication in Membranes after addressing the following concerns.
1. Authors presented figure 8 and 9. They also should be mentioned in the main text.
2. Figure 12 needs to be redrawn.
3. The selective layer of silver alginate plays an important role for the separation process. I am curious about the exact form of the silver in this layer. Can author explain that.
4. Authors need to discuss the separation mechanism of the prepared composite layer to guide readers to understand the separation process.
5. There are some mistakes in the main text. For example, line 201 “The exposure time to the crosslinking agent – 0.7 Ðœ 0.7 Ðœ solution silver nitrate”.
Author Response
Рецензент 1
Thank you for the review! All comments have been taken into account, which improved the proposed article for publication. All corrections in the text of the new version of the article are highlighted.
- Authors presented figure 8 and 9. They also should be mentioned in the main text.
Thank you for your response. Mentions of Figures 8 and 9 are pasted into the text (page 12).
- Figure 12 needs to be redrawn.
Thank you for your response. The missing marker was added to the drawing. In addition, when calculating the permeabilities, a mistake was made, and the numerical data were presented in other units, without taking into account the pressure. Therefore, the figure has been completely changed.
- The selective layer of silver alginate plays an important role for the separation process. I am curious about the exact form of the silver in this layer. Can author explain that.
The form of silver in membranes composition plays a major role in the separation of olefins and paraffins. Section 3.1 discusses «cross-linking» of alginate by the silver. It shows that according to the EDX data, crosslinking occurs throughout the membrane thickness (Figure 2) and more silver is introduced into the polymer than it would be in case with stoichiometric interaction of alginate and silver cations. This is made possible by the formation of silver nanoparticles in the polymer, which is confirmed by visual examination (darkening of the membranes) and spectral data (Figure 4). These nanoparticles interact with alginate due to the attraction of opposite charges (Figure 5).
However, it is worth noting that the darkening of membranes and the formation of silver nanoparticles occurs only in air and over time. Thus, initially the silver in the forming films - in the cross-linked alginate - is in cationic form. Part of this silver agglomerates over time, is reduced and transformed into nanoparticles. Part of the silver so and remains in cationic form, which is confirmed by experiments on their "washout" from dried membranes (in the previous version: page 9, lines 353 - 356).
Thus, in the cross-linked polymer silver is in two forms: silver cations, silver nanoparticles. Both of these forms are selective in the separation of olefins and paraffins. In order to make this thought better traceable in the article, lines 353-360 (in the former version) were moved and placed after Figure 5. The following sentence has also been added: " Both of those Ag forms (nanoparticles and cation) shows activity and selectivity in the separation of olefins and paraffins [81, 82]." (page 10, lines 364-365).
- Authors need to discuss the separation mechanism of the prepared composite layer to guide readers to understand the separation process.
The process of making selective layers on flat and hollow fiber composite membranes is presented in sections 2.3 and 2.5, respectively. This selective layer is an alginate with silver cations and nanoparticles distributed in it, as described in section 3.1. Silver in both forms is selective in the separation of olefins and paraffins and provides facilitated transport of unsaturated hydrocarbons through the polymer
- There are some mistakes in the main text. For example, line 201 “The exposure time to the crosslinking agent – 0.7 Ðœ 0.7 Ðœ solution silver nitrate”.
Thank you for your response. This misprint has been corrected.
Reviewer 2 Report
This work investigated the feasibility of an alginate membrane crosslinked by silver for ethylene/ethane separation. They also attempted to fabricate the corresponding composite membrane in flat-sheet and hollow fiber forms. This topic is of great interest to the readers of the “Membranes” Journal, while some details may need to be clarified before publishing. I would recommend a major revision for this manuscript, with comments listed below:
1. How do monovalent silver ions crosslink the alginate?
2. In Line 39, “Membrane separation processes are considered both as independent methods and working in tandem with distillation processes [8]. The latter approach allows a 30% reduction in energy consumption for separation [9]”. Does “the latter approach” refer to the “membrane separation processes”? If yes, it should be the “former one”, right?
3. Could the authors list some industrial “other olefin production processes”, in Line 60? To the reviewer’s knowledge, steam cracking is the primary production method. Probably more than 90% of ethylene is produced in this way.
4. Based on the definition of permeance, which is pressure normalized flux, equation 5 should be P=m/s/∆t/∆p, not P=m/s*∆t*∆p
5. In the second paragraph on Page 9, are “Figure 3A” and “Figure 3B” referring to “Figure 4”?
6. On page 10, the authors compared the sorption of silver alginate to calcium alginate, however, in the experimental part, the latter was not mentioned. Should the calcium alginate here be sodium alginate? If not, please provide more information about calcium alginate.
7. In line 376, “the ethylene/ethane vapor selectivity”? Both ethylene and ethane are gases under ambient conditions.
8. In section 3.3, the authors claimed the selective layer is defect-free because the water permeability of the dried membrane is 0. But later, extremely high gas permeances were reported for CO2, N2, and He (1 m3m-2h-2bar-1=365 GPU). Since water is smaller and more condensable than CO2, water should always be more permeable than CO2 in hydrophilic membranes. How does the silver alginate membrane block water but allow CO2 to pass through?
9. Can authors explain why the silver alginate membranes have such high gas permeance (> 1400 GPU for CO2 at ∆p of 35kPa)? Considering the thickness of the selective layer (>= 2.3 um), the corresponding CO2 permeability is higher than 3300 Barrer. But alginate is not famous for its high gas permeability (in contact actually). Also, why the CO2/N2 selectivity is so low (~ 2)?
10. In line 445, what does the “molecule size” high refer to? in the gas separation field, molecule size is generally described by the kinetic size, and CO2 has a smaller kinetic size (3.3 Å) than N2 (3.6 Å).
11. Please report the gas transport properties of ethane and ethylene in permeance, not permeability in Table 1.
12. How thick the selective layer of the membranes used for the permeation test is?
Author Response
Thank you for your detailed review of our article. All comments have been taken into account, which improved the proposed article for publication. All corrections in the text of the new version of the article are highlighted.
This work investigated the feasibility of an alginate membrane crosslinked by silver for ethylene/ethane separation. They also attempted to fabricate the corresponding composite membrane in flat-sheet and hollow fiber forms. This topic is of great interest to the readers of the “Membranes” Journal, while some details may need to be clarified before publishing. I would recommend a major revision for this manuscript, with comments listed below:
- How do monovalent silver ions crosslink the alginate?
The expression "cross-linked polymer" in this case does not mean a specific mechanism of interaction of substances and, for example, the formation of bridging ionic or covalent bonds, as in the case of cross-linking of alginate by calcium cations (Hecht, H., & Srebnik, S. (2016). Structural characterization of sodium alginate and calcium alginate. Biomacromolecules, 17(6), 2160-2167.) or glutaric aldehyde (Wang, Q., Ju, J., Tan, Y., Hao, L., Ma, Y., Wu, Y., . & Sui, K. (2019). Controlled synthesis of sodium alginate electrospun nanofiber membranes for multi-occasion adsorption and separation of methylene blue. Carbohydrate Polymers, 205, 125-134.), and the transition process of alginate from the water-soluble sodium form to the insoluble polymer form. Silver in alginate is in two states: in the form of nanoparticles and in the form of cations. Both of these forms form bonds with alginate by ionic and coordination mechanisms, which has already been proven in the literature (page 3, lines 110-116). This is due to the charge difference between the polymer and silver cations, as well as to the presence of free d-orbitals in silver, capable of "settling" the unseparated oxygen electron pairs from hydroxyl and carboxyl groups
- In Line 39, “Membrane separation processes are considered both as independent methods and working in tandem with distillation processes [8]. The latter approach allows a 30% reduction in energy consumption for separation [9]”. Does “the latter approach” refer to the “membrane separation processes”? If yes, it should be the “former one”, right?
The "latter approach" is a combination of distillation and membrane separation. This approach reduces costs by up to 30 % compared to the distillation approach. The combination of processes is the best way to realize separation processes. The use of "pure" traditional processes is associated with high energy and material costs. The use of "pure" membrane processes is associated with a high load on the membrane modules, the need for their frequent replacement, lower performance and other
- Could the authors list some industrial “other olefin production processes”, in Line 60? To the reviewer’s knowledge, steam cracking is the primary production method. Probably more than 90% of ethylene is produced in this way.
The authors meant that the humidity of the feed stream is important for the use of membranes with the inclusion of silver ions. The controversial phrase was removed from the text of the article
- Based on the definition of permeance, which is pressure normalized flux, equation 5 should be P=m/s/∆t/∆p, not P=m/s*∆t*∆p
Thank you. Formula the formula has been corrected.
- In the second paragraph on Page 9, are “Figure 3A” and “Figure 3B” referring to “Figure 4”?
Thank you for your response. Of course, instead of Figures 3A and 3B there should be references to Figures 4A and 4B. The references to the figures have been corrected
- On page 10, the authors compared the sorption of silver alginate to calcium alginate, however, in the experimental part, the latter was not mentioned. Should the calcium alginate here be sodium alginate? If not, please provide more information about calcium alginate.
The sorption for ethane and ethylene was taken for the original polymer - sodium alginate, and for alginate cross-linked with silver. The text and the figure has been corrected in the revised version of the article
- In line 376, “the ethylene/ethane vapor selectivity”? Both ethylene and ethane are gases under ambient conditions.
Thank you for the remark. Ethylene and ethane are gases, not vapors. The extra word "vapor" has been removed from the text
- In section 3.3, the authors claimed the selective layer is defect-free because the water permeability of the dried membrane is 0. But later, extremely high gas permeances were reported for CO2, N2, and He (1 m3m-2h-2bar-1=365 GPU). Since water is smaller and more condensable than CO2, water should always be more permeable than CO2in hydrophilic membranes. How does the silver alginate membrane block water but allow CO2 to pass through?
Thank you for your response. Silver alginate has no ability to block water. Under experimental conditions at the minimum pressure drop (0.2 atm), there was no visible water flow through the membrane. Thus, we confirmed the absence of defects in the formation of the selective layer. The term "water permeability" was replaced by "water flux," which means the flow of liquid water through the membrane. The incorrect wording has been replaced in the text of the article.
The gas permeability for these membranes was recalculated and proved to be significantly lower than the previously presented values (Table 1).
- Can authors explain why the silver alginate membranes have such high gas permeance (> 1400 GPU for CO2at ∆p of 35kPa)? Considering the thickness of the selective layer (>= 2.3 um), the corresponding CO2 permeability is higher than 3300 Barrer. But alginate is not famous for its high gas permeability (in contact actually). Also, why the CO2/N2 selectivity is so low (~ 2)?
Thank you for the valuable remark! The data were recalculated (normalized on the pressure drop) and converted to GPU units (Table 1, Figure 12).
The found value of selectivity was reproduced on a series of both flat and composite type membranes. Such value seems low for alginic acid, however it is known [T.Anokhina, E.Dmitrieva, A.Volkov. Recovery of Model Pharmaceutical Compounds from Water and Organic Solutions with Alginate-Based Composite Membranes. Membranes, 12 no. 2 (2022) 235] that the properties of alginate-based membranes strongly depend on the cation. No gas permeability data have been previously presented for silver alginate. The distribution of gas permeability by their kinetic diameters correlates well with transport by the dissolution-diffusion mechanism (which describes transport through non-porous layers)
- In line 445, what does the “molecule size” high refer to? in the gas separation field, molecule size is generally described by the kinetic size, and CO2has a smaller kinetic size (3.3 Å) than N2 (3.6 Å).
Thank you for the remark. This text was written using data from an incorrect source (http://www.sci.aha.ru/ALL/b4.htm). Due to data on the kinetic size of molecules, this paragraph has been rewritten.
- Please report the gas transport properties of ethane and ethylene in permeance, not permeability in Table 1.
The data in Table 1 have been converted from Barrer units to GPUs. In this regard, the numerical data in the conclusion are also changed.
- How thick the selective layer of the membranes used for the permeation test is?
The thickness of the selective layer of silver alginate on the flat composite membrane was 20 µm. On the hollow fiber, the alginate was applied once from a 3% solution and was crosslinked for 5 seconds, corresponding to 8.3 μm (Figure 8). Information about this is added in the caption of Figure 12 and Table 1.
Round 2
Reviewer 2 Report
The current manuscript is qualified to be published in the Membranes.